

# Comprehensive proteomic and metabolomic analysis uncover the response of okra to drought stress

Jiyue Wang, Denghong Shi, Yu Bai, Ting Zhang, Yan Wu, Zhenghong Liu, Lian Jiang, Lin Ye, Zele Peng, Hui Yuan and Yan Liu

Guiyang University, Guiyang, China

## ABSTRACT

The response of okra to drought stress is very complicated, and the molecular mechanisms underlying this process remains ambiguous up to now. In this study, different degrees of water-stress responses of okra leaf were explained by using transcriptomics and metabolomic approaches. The photosynthesis and glycometabolism in okra leaf were both adversely affected by drought stress, leading to inhibition of the carbohydrate metabolic process, and then influencing the secondary plant metabolism. Further, drought stress disturbed amino acid metabolism, especially for the tyrosine-derived pathway as well as arginine and proline metabolism, which have been shown to be significantly enriched under water withholding conditions based on multi-omics conjoint analysis (transcriptome, proteome and metabolome). In-depth analysis of the internal linkages between differentially expressed transcripts, proteins, and metabolites decidedly indicate that tyrosine metabolism could confer tolerance to drought stress by influencing carbon and nitrogen metabolism. These findings provide a whole framework of the regulation and relationships of major transcripts and peptides related to secondary metabolism, particularly, the role of critical proteins and metabolite involved in the change of amino acid metabolism in response to drought stress.

## INTRODUCTION

Okra (*Abelmoschus esculentus* L. Moench), which belongs to the Malvaceae family, originated in Africa and India and is able to adapt to a wide range of warm climates (*Gemede et al., 2016*). Okra is an important, healthy vegetable and is very popular in various parts of the world. The value of one ton of okra varies worldwide, with 2017 prices ranging from $236.8 USD in Mexico to $3,870.6 USD in Fiji. A series of studies have shown that okra polysaccharide could be used as a potential immunomodulator for the treatment of diabetic nephropathy (*Chen et al., 2016*; *Peng et al., 2016*). The rhamnogalacturonan polysaccharide found in okra is also associated with hypoglycemic effects (*Liu et al., 2018*).

The growth and development of plants is often compromised by abiotic stresses such as drought. Plants undergo substantial changes in their physiological and biochemical

Corresponding authors
Jiyue Wang, acute2803764@126.com
Yan Liu, Liuyan19680607@126.com

systems when faced with water deficiency (*Farias et al., 2019*). Recent studies have shown that under drought conditions, the biomass of okra as well as the uptake of phosphorus in its shoot were both significantly reduced, while nitrogen, potassium, iron, and zinc levels increased in the shoot (*Müller, Eltigani & George, 2019*). Water deficits affect the physiology and development of okra, and severe water shortages can significantly reduce okra production. Improving okra irrigation techniques and cultivating a new drought-resistant variety of okra are two effective ways to solve this problem. *Amin et al. (2009)* found that a 1 mM concentration of salicylic acid and ascorbic acid can considerably mitigate the physical damage to plants caused by drought stress.

In recent years, the rise of omics studies has provided an important means of revealing the response of plants to biotic or abiotic stress. The molecular mechanisms underlying OsDRAP1-mediated salt tolerance in rice was revealed through comparative transcriptome and metabolome analyses (*Wang et al., 2021*). *Li et al. (2021)* reported the mechanisms at play in the molecular and physiological metabolic response of *N. sibirica* to salt stress by using comprehensive transcriptome and metabolome profiling. *Bavaresco et al. (2020)* found that protein hydrolysates modulate the leaf proteome and metabolome of grapevines in response to water stress. The mechanism of Se accumulation and tolerance in *C. violifolia* was also identified using metabolome, transcriptome, and proteome technologies (*Rao et al., 2021*). The protein turnover and regulatory classes of proteins and metabolites in *Medicago truncatula* during drought stress and subsequent recovery were identified through an integration of proteome and metabolome analyses (*Lyon et al., 2016*).

Previous studies on okra have focused mainly on the characterization of its genotypes (*Ghevariya & Mahatma, 2017*), its medical applications (*Erfani et al., 2018*), its agronomic characteristics (*Meldrum et al., 2018*), and its edible quality (*Petropoulos et al., 2018*). However, few studies report on the molecular mechanism of resistance to drought stress in okra plants. The aim of this study is to reveal the drought-resistant mechanism of okra at the molecular level. The protein expression and metabolic profiles of okra under different water withholding conditions were obtained using a multi-omics analysis. The functional proteins and metabolites associated with drought tolerance, and the metabolic pathways involved were also identified using a proteomic analysis and a metabolomics analysis, respectively.

## MATERIALS AND METHODS

### Plant materials

A drought-tolerant okra cultivar called 'Xianzhi' was selected based on a previous physiological and biochemical experiment (*Wang et al., 2018b*). It was then cultivated in a greenhouse at the Guiyang University in Guizhou province, China. Blades from seedling cuttings were used to extract total proteins and metabolites.

### Drought stress treatment

Drought treatment was carried out in a constant temperature incubator. Okra plants were planted in plastic buckets 20.0 cm in height with a 15.0 cm inner diameter. They were cultivated for 35 days at 70% humidity and a temperature of 25 ± 2 °C. First, the okra

plants were planted by direct seeding into a basin containing nutrient-enriched soil. After sprouting, the seedlings were watered every 2 days. Then, after an adaptation period of 2 weeks, a dehydration treatment was applied to all plants. Leaves were collected from the seedlings after 0 days, 5 days, 7 days, 15 days, and 20 days of water withholding. The collected leaves were then kept in liquid nitrogen for protein extraction and stored at −80 °C in an ultra-low temperature refrigerator. The five different drought treatments were marked as P1, P2, P3, P4, and P5, respectively, with samples taken from each drought treatment. Using a randomized block design, 11 pots were used in each treatment with three used for proteomic assays and the other eight for metabolomic analysis.

## Sample processing and TMT quantification

The protein was extracted using the methods described by *Xiong et al. (2019)*. After trypsin digestion (where a protease inhibitor was added at a rate of 50:1), 8 M urea was added and an ultrasound was performed for 1 s, and then stopped for 2 s, with that pattern repeated for a total of 20 s. After centrifugation at 14,000 g for 20 min, 5 μL of the supernatant was kept for quantification, and the rest was frozen at −80 °C. The protein concentration was determined using the Bradford method. SDS-page was performed using 20 μg of each sample with Coomassie blue staining for 30 min followed by decolorization until the background was clear. FASP (Filter Aided Sample Preparation) was then carried out using a TMT® kit (Thermo Fisher Scientific, Waltham, MA, USA). After enzymatic digestion, 41 μL of TMT reagent was added to a 100 μg sample (100 μL per sample), and incubated at room temperature for 1 h. Then, 8 μL of 5% quenching reagent (Thermo Fisher Scientific, Waltham, MA, USA) was added and incubated for 15 min to stop the reaction. The mixed and labeled samples were centrifuged to the bottom of the tube by vortex, and then dried with centrifugal vacuum freezing.

## Peptide pre-separation and LC-MS/MS analysis

The tryptic peptides were dissolved in solvent A (2% acetonitrile, PH 10) to 100 μL, then centrifuged at 14,000 g for 20 min, and the supernatant was removed and put into a custom-made reverse-phase analytical column (Durashell-C18, 4.6 mm × 250 mm, 5 μm, 100 A). It took 5 min for solvent B to move from 5% to 8% (98% acetonitrile, PH 10), an additional 30 min for it to grow from 8% to 18%, another 27 min for it to reach 32%, and then just 2 min for it to move from 32% to 95%. The 95% held for 4 min and then decreased all the way to 5% in the next 4 min, all at a constant flow rate of 0.7 ml/min on an RIGOL L-3000 high performance liquid chromatography system (Beijing Puyuan Jing Electric Technology Co., LTD, Qingdao, China).

The components obtained from high pH reversed phase separation were redissolved in reagent with 2% methanol and 0.1% formic acid, centrifuged at 12,000 g for 10 min, and then the supernatant was loaded onto an EASY-Spray column (12 cm × 75 μm, C18, 3 μm). The loading pump was running for 15 min at a flow rate of 350 nL/min. Peptides were separated using the EASY-nLC 1,000 System (Nano HPLC, Thermo Fisher Scientific, Waltham, MA, USA) at a constant flow rate of 600 nL/min. The separation gradient is shown in Table 1.
**Table 1 Separation gradient.**

| Time (min) | Mobile phase A (0.1%FA/H$_2$O) | Mobile phase B (0.1%FA/ACN) |
|---|---|---|
| 0 | 93% | 7% |
| 11 | 85% | 15% |
| 48 | 75% | 25% |
| 68 | 60% | 40% |
| 69 | 0% | 100% |
| 75 | 0% | 100% |

The peptides were then injected into an NSI ion source for ionization and analyzed using Orbitrap Fusion Lumos (Thermo Fisher Scientific, Waltham, MA, USA) mass spectrometry. The ion source voltage was set to 2.0 kV, and the capillary temperature was 320 °C. The mass spectrometer scan range was set to 300–1,400 m/z, and the scan resolution was set to 120,000 FWHM. The full scan automatic gain control (AGC) target, and full scan Max.IT (maximum implantation time) were set to 5.0e$^5$ and 50 ms, respectively. The dd-MS2 resolution was set to 60,000 FWHM and 35% fragmentation energy was used for fragmentation according to the higher energy collision dissociation (HCD) method. The automatic gain control (AGC) target was set to 5.0e$^4$, and the Max.IT was set to 118 ms.

The resulting MS/MS data were then analyzed using the Proteome Discoverer (v.2.1). The tandem mass spectra were searched against the *Abelmoschus esculentus* L. corresponding transcriptome database (*Shi et al., 2020*) and the UniProt/NCBI database. The enzyme digestion method was set as trypsin; the max missed cleavages was set as 2; the tolerances of precursor ion mass and fragment ion mass were set as 15 ppm and 20 ppm, respectively; the static modification and dynamic modification were set as C carboxyamidomethylation (57.021 Da) and M oxidation (15.995 Da), respectively; and the quantitative method was set as iTRAQ-6plex.

## Peptide identification and differentially expressed protein (DEP) screening

The peptides produced through the enzymatic hydrolysis of the proteins were identified through mass spectrometry, and then the putative protein was obtained using a bioinformatics analysis. In order to evaluate the overall picture of the proteomic data, the physical and chemical properties were detected at both the peptide and protein levels. For peptides, this meant calculating: peptide length, PSM number distribution, score distribution for identified peptides, and missed cleavage distribution for identified peptides. For proteins: distribution of identified peptide numbers for proteins, distribution of PSM numbers matched to proteins, MW distribution for identified proteins, coverage distribution for identified proteins, and pI distribution of identified proteins were all calculated.

Since the sample was repeated ≥2 times, a t-test was used for differential analysis. DEPs were defined with a *P*-value of <0.05, and a fold change (FC) of >1.2 between any two treatments.

## Functional annotation of proteins

The Clusters of Orthologous Groups (COG) analysis was achieved by blasting KYVA sequences. The Gene Ontology (GO) and The Kyoto Encyclopedia of Genes and Genomes (KEGG) annotations were acquired using *Arabidopsis Thaliana* annotated data in Uniprot. The PPI (Protein-Protein Interaction Networks) analysis used *Arabidopsis Thaliana* data in the STRING database to search the relationship between the DEPs and their possible functional groups.

An enrichment analysis was used to determine the over-expressed genes or proteins, allowing further analyses to identify the functional categories or pathways involved. An over-representation analysis was used to perform a statistical significance test according to hypergeometric distribution. The *P* values and false discovery rate (FDR) values (based on multiple hypothesis testing) of the enrichment degree from differential proteins were calculated based on the functional categories of GO and Go Slim as well as the KEGG pathways; the smaller the P value or FDR value, the higher the enrichment degree.

The GO analysis was scattered, and it was difficult to draw overall conclusions based on the overly complex and detailed classification results. However, GO Slim is a simplified version of GO, which matches most entries to a few parent entries, making it easy to obtain the protein number and enrichment degree contained in each large entry. Like the GO analysis, the GO Slim annotation is divided into three major categories: biological process (BP), cellular component (CC), and molecular function (MF).

## Protein-protein interaction (PPI) analysis

The identified DEPs were used to construct the PPI network to explore the inter-class relationships and possible functional groups of the DEPs. STRING (Search Tool for the Retrieval of Interacting Genes/Proteins) is part of the Elixir infrastructure, and is one of Elixir's core data resources. The DEPs were uploaded to the STRING 11.0 database (https://string-db.org/), and the interacting proteins were identified based on *Arabidopsis thaliana* as the model organism. Protein-protein interactions were identified using a combined score of 0.4 as the threshold. The Cytoscape 3.6.1 software (*Shannon et al., 2003*) was used to visually construct the protein interaction network.

## Metabolite extraction

The metabolites were extracted according to the De Vos RC1 method (*De Vos et al., 2007*) and the approach of *Sangster et al. (2006)*. Each of the treatments, containing six replicates, were used for this metabolomic analysis. After weighing, 200 mg (±2%) of each sample was put in a 2 mL EP tube, 0.6 mL 2-chlorophenylalanine (4 ppm) methanol (−20 °C) was added, and then the sample was vortexed for 30 s. After that, 100 mg glass beads were added to each sample and the samples were put into the TissueLysis II tissue grinding

machine and ground at 25 Hz for 60 s, followed by an ultrasound at room temperature for 15 min. The samples were then centrifuged at 25 °C for 10 min at 1,750 g, and the supernatant was filtered through a 0.22 μm membrane to obtain the samples necessary for LC-MS. A 20 μL quality control sample was taken from each sample (Fig. S1) and used to monitor deviations in the analytical results from the pool mixtures and compare them to the errors caused by the analytical instrument itself. The remaining part of each sample was used for LC-MS detection.

## LC-MS analysis

Chromatographic separation was completed with a ThermoUltimate 3,000 system equipped with an ACQUITY UPLC® HSS T3 (150 × 2.1 mm, 1.8 μm, water) column maintained at 40 °C. The temperature of the autosampler was set to 8 °C. Gradient elution of analytes was implemented with 0.1% formic acid in water (C) and 0.1% formic acid in acetonitrile (D) or 5 mM ammonium formation water (A) and acetonitrile (B) at a flow rate of 0.25 mL/min with 2 μL of each sample injected after equilibration. An increasing linear gradient of solvent B (v/v) was carried out as specified in the manufacturer's instructions.

The ESI-MSn experiments were performed on the Thermo Q Exactive mass spectrometer with a spray voltage of 3.8 kV and −2.5 kV in positive and negative modes, respectively. Auxiliary gas and sheath gas were set at 10 and 30 arbitrary units, respectively. The capillary temperature was set at 325 °C. The analyzer scanned over a mass range of $m/z$ 81-1,000 for full scan at a mass resolution of 70,000. The data dependent acquisition of the MS/MS spectra was carried out using an HCD scan. The normalized collision energy was set to 30 eV. Dynamic exclusion was used to remove unnecessary information from the MS/MS spectra. The original data obtained by the ProteWizard software (V3.08789) was converted into the mzXML format. The XCMS program of R was used to carry out peak identification, peak filtration, and peak alignment, leading to the building of the data matrix including mass to change ratio (m/z), retention time (r/t), and intensity.

The original LC-MS data of the metabolites were standardised and used for the principal component analysis (PCA), partial least squares discriminant analysis (PLA-DA), and orthogonal projections to latent structures-discriminant analysis (OPLA-DA).

The metabolomics profiles were investigated as described by *Zhong et al. (2022)*. Differential metabolites (DMs) were identified according to $P$-value ($P > 0.05$) from a two-tailed Student's t-test on the normalised peak areas. The pheatmap program package in R (V3.3.2) was used to carry out agglomerative hierarchical clustering. A pathway enrichment analysis was carried out using the KEGG database with $P$-values $<0.05$ considered a significant enriched pathway. A correlation analysis of the differential metabolites was also performed in this study.

# RESULTS

## Protein identification and evaluation

A total of 18,875 peptides aligning to 4,151 proteins were identified by means of TMT analysis. The results were highly reliable in detecting the physiological-biochemical
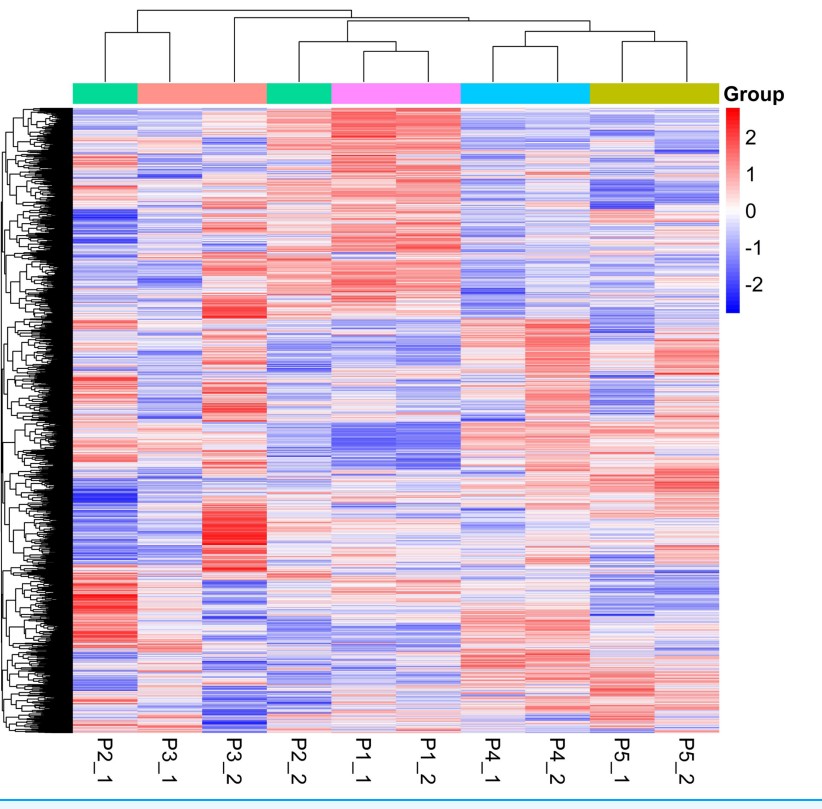

**Figure 1** **Global heatmap.**               

properties of the identified peptide and its presumptive protein (Figs. S2–10), which could then be used for subsequent analysis.

## DEP identification and functional description

Quantitative values of different labels in the PD search results were directly extracted, and the global view of the DEPs (Fig. 1) was obtained after removing the results with 0 value. DEPs were identified through pairwise comparison between the different treatments (after 0 days, 5 days, 7 days, 15 days, and 20 days of water withholding). Ten sample pairs and the number of DEPs identified in the pairwise comparison of each sample pair are shown in Fig. 2: P2 *versus* P1 (126 DEPs), P3 *versus* P1 (363 DEPs), P4 *versus* P1 (1,015 DEPs), P5 *versus* P1 (791 DEPs), P3 *versus* P2 (46 DEPs), P4 *versus* P2 (245 DEPs), P5 *versus* P2 (261 DEPs), P4 *versus* P3 (170 DEPs), P5 *versus* P3 (159 DEPs), and P5 *versus* P4 (236 DEPs). Most of the DEPs identified were shared among the ten pairs. In particular, the number of DEPs found between each treatment and the control (P1) first increased and then decreased with increased levels of water stress. The number of DEPs between 15 days of dehydration (P4) and the control (P1) was the most abundant, followed by P5 *versus* P1, and then P3 *versus* P1. However, the number of DEPs found among the different drought treatments were less the number found between each treatment and control. For example, a total of 363 DEPs were found in P3 *versus* P1, but only 46 DEGs were found in P3 *versus* P2. Moreover, the number of down-regulated DEPs found between each treatment and control was higher than between the different treatments. More up-regulated DEPs were

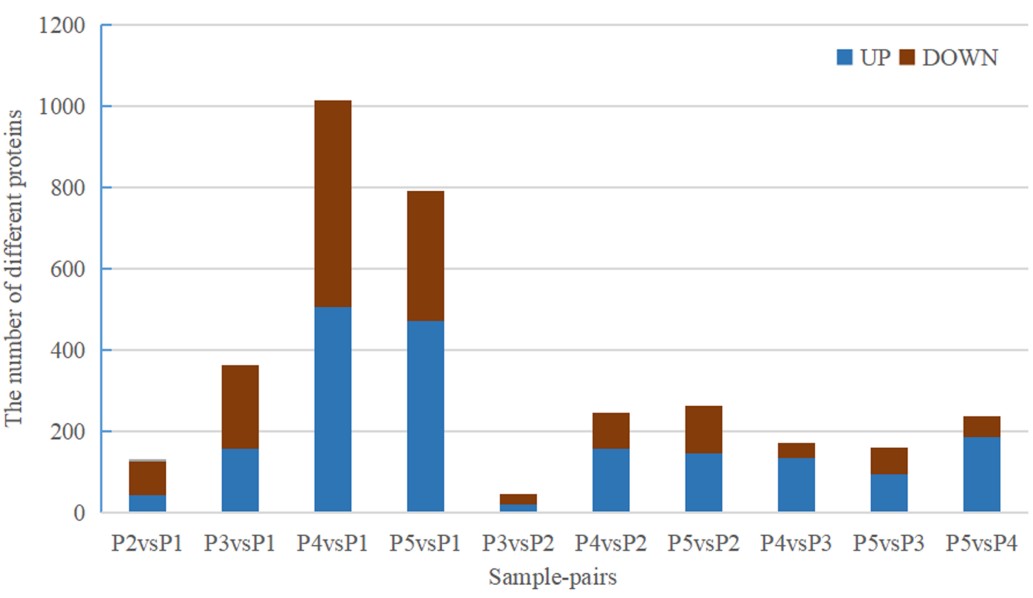

**Figure 2** Statistics of DEPs from ten sample-pairs, being namely P2 *versus* P1, P3 *versus* P1, P3 *versus* P2, P4 *versus* P1, P4 *versus* P2, P4 *versus* P3, P5 *versus* P1, P5 *versus* P2, P5 *versus* P3, and P5 *versus* P4. Red: upregulated expressed proteins, blue: down-reg.

found in P4 *versus* P2, P5 *versus* P2, P4 *versus* P3, P5 *versus* P3, and P5 *versus* P4 than were found in P3 *versus* P2.

All the differentially expressed proteins that were identified were annotated by aligning them to the COG database, which identifies lineal homologous genes through an extensive comparison of protein sequences from a wide variety of organisms. A total of 2,818 DEPs were grouped into 25 COG categories (Fig. 3). The largest category was "general function prediction only" containing 406 DEGs (14.41%), followed by "posttranslational modification, protein turnover, chaperones" (393 DEGs, 13.95%), "translation, ribosomal structure and biogenesis" (259 DEGs, 9.19%), "energy production and conversion" (221 DEGs, 7.8%), and "carbohydrate transport and metabolism" (205 DEGs, 7.27%). Only four DEPs were assigned to the "cell motility" category and eight to the "extracellular structures" category. In addition, there were 70 DEPs assigned to "function unknown," accounting for 2.24%.

### GO analysis of DEPs

According to the results of the enrichment analysis, all DEPs in the ten pairs (P2 *versus* P1, P3 *versus* P1, P3 *versus* P2, P4 *versus* P1, P4 *versus* P2, P4 *versus* P3, P5 *versus* P1, P5 *versus* P2, P5 *versus* P3, and P5 *versus* P4) were classified into 1,809 subgroups of biological process (BP), 438 subgroups of cellular component (CC), and 1,341 subgroups of molecular function (MF). Fig. 4 shows the top 20 GO categories of all DEPs. The "response to cadmium ion" was the largest BP category, involving 286 DEPs. There were 1,225 DEPs involved in "chloroplast," which was the largest CC category, and the "structural constituent of ribosome" was largest MF category, including 243 DEPs. Furthermore, all DEPs were involved in 17 cellular component categories, seven molecular function

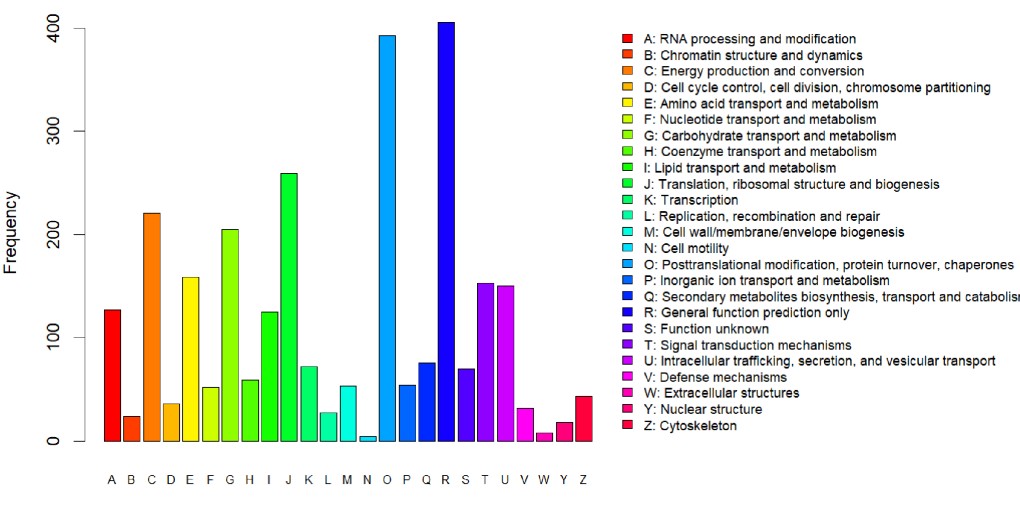

**Figure 3 COG annotation.**

categories, and 20 biological process categories through a GO Slim analysis. In this analysis, the largest BP category was "transport" (420 DEPs), the largest CC category was "cytoplasm" (3,025 DEPs), and the largest MF category was "metal ion binding" (875 DEPs; Fig. 5). The GO Slim analysis of DEPs was also performed in the ten pairs (P2 *versus* P1, P3 *versus* P1, P3 *versus* P2, P4 *versus* P1, P4 *versus* P2, P4 *versus* P3, P5 *versus* P1, P5 *versus* P2, P5 *versus* P3, and P5 *versus* P4) with "metal ion binding" as the largest MF category for all ten sample pairs. There were 27, 73, 11, 238, 57, 37, 181, 71, 39, and 46 DEPs enriched in "metal ion binding" in P2 *versus* P1, P3 *versus* P1, P3 *versus* P2, P4 *versus* P1, P4 *versus* P2, P4 *versus* P3, P5 *versus* P1, P5 *versus* P2, P5 *versus* P3, and P5 *versus* P4, respectively. The largest CC category was also "cytoplasm," with 100, 259, 39, 752, 178, 128, 597, 199, 120, and 177 DEPs enriched in each of the ten pairs, respectively. The largest BP category differed by pair: "carbohydrate metabolic process" was the largest in P3 *versus* P1 (30), P4 *versus* P1 (93), P5 *versus* P1 (75), P3 *versus* P2 (6), P4 *versus* P2 (54), P5 *versus* P2 (26), P4 *versus* P3 (15), and P5 *versus* P4 (22); "signal transduction" was the largest in P2 *versus* P1 (19); and "transport/Reproduction" was the largest in P5 *versus* P3 (13). More down-regulated GO terms were found in P2 *versus* P1, P3 *versus* P1, P3 *versus* P2, and P5 *versus* P1, with the other pairs having more up-regulated GO terms than down-regulated.

## KEGG pathway of DEPs

The KEGG enrichment analysis of the DEPs showed that 125 metabolic pathways were obtained from all DEPs, 37 of which were significantly (*P* < 0.01) enriched KEGG pathways (Table 2). There were seven pathways with enrichment values >10: "metabolic pathways," "carbon metabolism," "carbon fixation in photosynthetic organisms," "biosynthesis of amino acids," "biosynthesis of secondary metabolites," "photosynthesis," and "pyruvate metabolism" (Table S1). In addition, 50 KEGG pathways were enriched in

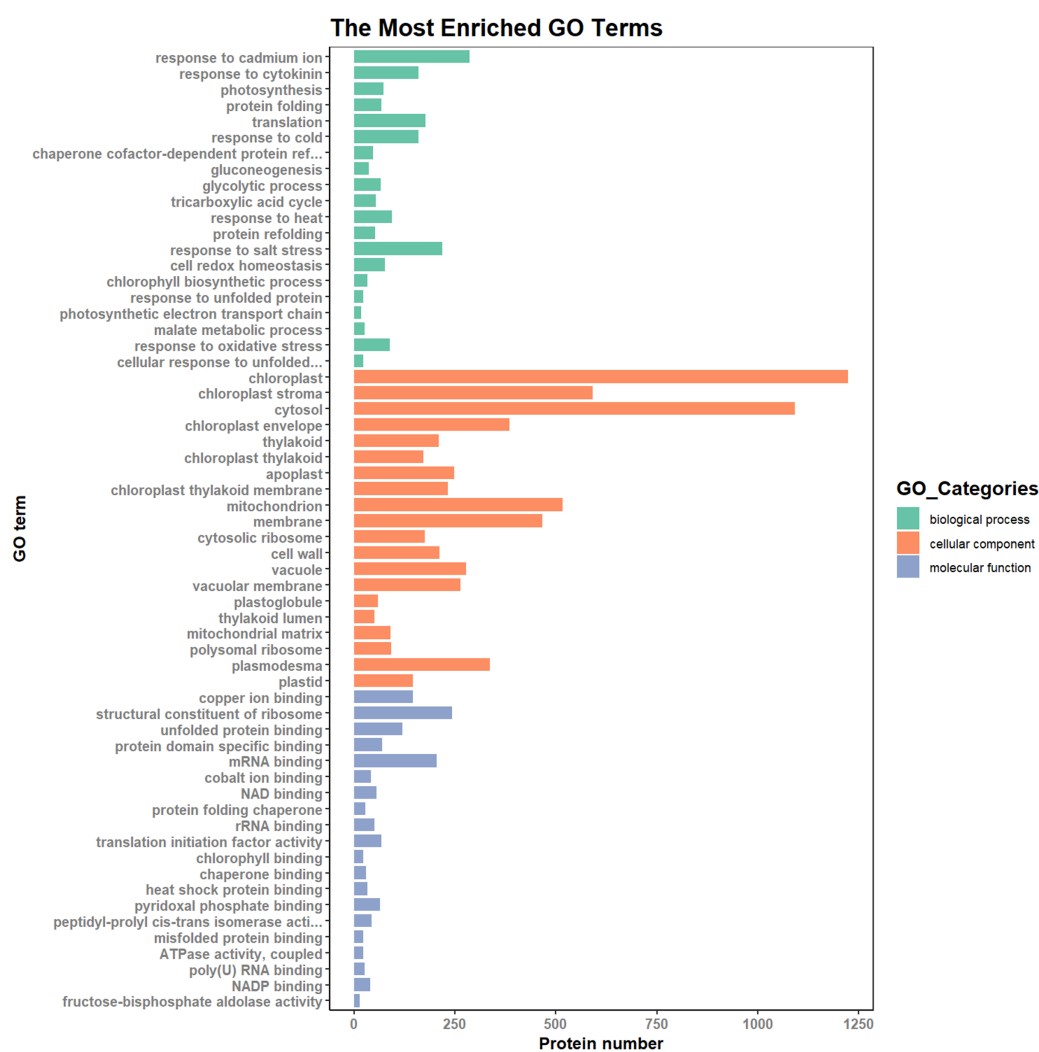

**Figure 4 Top 20 GO categories of all DEPs.**

P2 *versus* P1, 81 in P3 *versus* P1, 27 in P3 *versus* P2, 110 in P4 *versus* P1, 79 in P4 *versus* P2, 71 in P4 *versus* P3, 102 in P5 *versus* P1, 79 in P5 *versus* P2, 57 in P5 *versus* P3, and 65 in P5 *versus* P4, as shown in Figs. S21–S30. Among the ten sample pairs, the top three enriched pathways were "metabolic pathways," "biosynthesis of secondary metabolites," and "biosynthesis of amino acids." There were more down-regulated DEP pathways than up-regulated in P2 *versus* P1, P3 *versus* P1, P3 *versus* P2, and P4 *versus* P1, and more up-regulated DEP pathways in the rest of the sample pairs (Table S2). Interestingly, up-regulated DEP pathways linked to "protein processing in endoplasmic reticulum" were found in P4 *versus* P3, and in P5 *versus* P3 (Figs. S31–S32), while "ribosom" was the predominant pathway of up-regulated DEPs in P2 *versus* P1, P3 *versus* P1, P4 *versus* P1, and P5 *versus* P1. The main pathways of the down-regulated DEPs in all ten sample pairs were "metabolic pathways" and "biosynthesis of secondary metabolites".

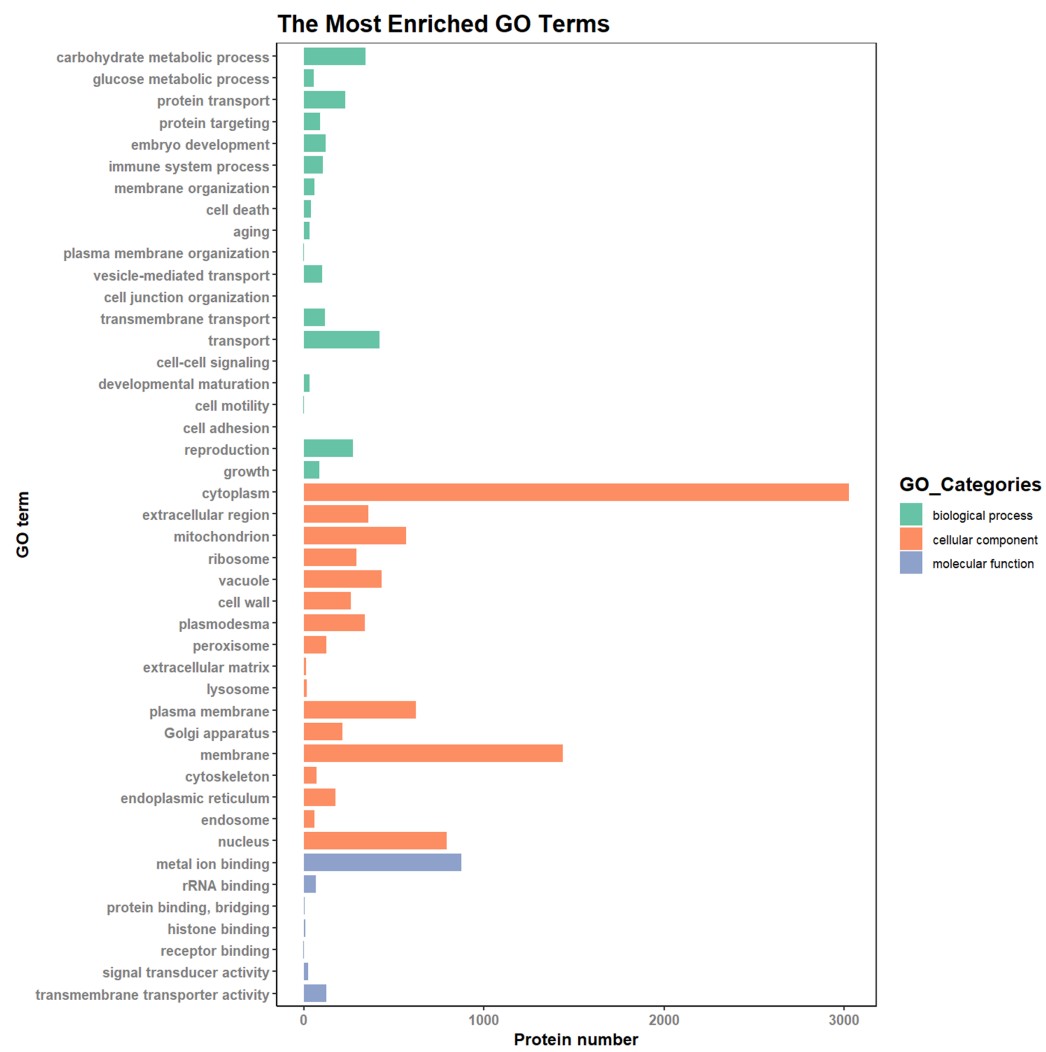

**Figure 5  The most enriched GO terms in GO Slim.**     

## PPI network for exploring hub proteins associated with drought stress responses in okra

In order to search potential proteins associated with drought stress responses, all DEPs in four pairs (P2 *versus* P1, P3 *versus* P1, P4 *versus* P1, P5 *versus* P1) were used to construct the PPI network. Four large networks with several smaller networks were obtained from the DEPs in these four sample pairs with 86 DEPs involved in protein interaction in P2 *versus* P1, 284 DEPs in P3 *versus* P1, 884 DEPs in P4 *versus* P1, and 679 DEPs in P5 *versus* P1. Among these interacting proteins, more down-regulated DEPs were detected in all of these pairs except in P4 *versus* P1.

The four large networks obtained from the P2 *versus* P1, P3 *versus* P1, P4 *versus* P1, and P5 *versus* P1 pairs are shown in Fig. 6. The large P2 *versus* P1 network contained 114 nodes linking 335 edges (Fig. 6A); the P3 *versus* P1 network comprised 313 nodes connecting 2,202 edges (Fig. 6B); the P4 *versus* P1 network had 828 nodes and 17,798 edges (Fig. 6C); and the P5 *versus* P1 network had 660 nodes linking 9,767 edges (Fig. 6D). The key nodes

**Table 2 The significantly (P < 0.01) enriched KEGG pathways.**

| Term ID | Term description | Termnum | P-value | Ratio | Enrichment | FDR |
|---|---|---|---|---|---|---|
| path:ath01100 | Metabolic pathways | 937 | 0.157346767 | 1.93E−26 | 25.7150997 | 2.41E−24 |
| path:ath01110 | Biosynthesis of secondary metabolites | 557 | 0.163391024 | 2.15E−16 | 15.66656637 | 5.39E−15 |
| path:ath01200 | Carbon metabolism | 238 | 0.234714004 | 2.88E−25 | 24.54112048 | 1.80E−23 |
| path:ath03010 | Ribosome | 203 | 0.169449082 | 3.25E−07 | 6.487789383 | 2.90E−06 |
| path:ath01230 | Biosynthesis of amino acids | 183 | 0.220481928 | 1.68E−16 | 15.77493564 | 5.25E−15 |
| path:ath03040 | Spliceosome | 103 | 0.161189358 | 0.001667046 | 2.778052308 | 0.007185545 |
| path:ath00010 | Glycolysis/Gluconeogenesis | 96 | 0.207343413 | 8.59E−08 | 7.065859234 | 9.76E−07 |
| path:ath00620 | Pyruvate metabolism | 95 | 0.246753247 | 6.53E−12 | 11.18495109 | 1.17E−10 |
| path:ath00710 | Carbon fixation in photosynthetic organisms | 85 | 0.307971014 | 1.15E−16 | 15.93857506 | 4.80E−15 |
| path:ath00190 | Oxidative phosphorylation | 77 | 0.168859649 | 0.001769333 | 2.752190339 | 0.007372222 |
| path:ath00630 | Glyoxylate and dicarboxylate metabolism | 73 | 0.253472222 | 5.13E−10 | 9.28984492 | 8.02E−09 |
| path:ath00020 | Citrate cycle (TCA cycle) | 63 | 0.259259259 | 3.03E−09 | 8.518991062 | 4.20E−08 |
| path:ath00260 | Glycine, serine and threonine metabolism | 60 | 0.256410256 | 1.12E−08 | 7.951195634 | 1.40E−07 |
| path:ath00270 | Cysteine and methionine metabolism | 59 | 0.184952978 | 0.000659883 | 3.180532956 | 0.003299416 |
| path:ath00195 | Photosynthesis | 58 | 0.364779874 | 2.14E−15 | 14.66871 | 4.47E−14 |
| path:ath00230 | Purine metabolism | 56 | 0.178913738 | 0.001965531 | 2.706520092 | 0.007925528 |
| path:ath00051 | Fructose and mannose metabolism | 54 | 0.215139442 | 1.93E−05 | 4.714206495 | 0.000127043 |
| path:ath00970 | Aminoacyl-tRNA biosynthesis | 52 | 0.254901961 | 1.29E−07 | 6.891076785 | 1.34E−06 |
| path:ath00480 | Glutathione metabolism | 52 | 0.228070175 | 4.85E−06 | 5.314302303 | 3.57E−05 |
| path:ath01210 | 2-Oxocarboxylic acid metabolism | 49 | 0.212121212 | 6.71E−05 | 4.173445821 | 0.000419212 |
| path:ath03050 | Proteasome | 49 | 0.210300429 | 8.41E−05 | 4.074974554 | 0.00050086 |
| path:ath01212 | Fatty acid metabolism | 49 | 0.17562724 | 0.005242324 | 2.280476165 | 0.019273249 |
| path:ath00250 | Alanine, aspartate and glutamate metabolism | 47 | 0.262569832 | 1.99E−07 | 6.701294506 | 1.91E−06 |
| path:ath00030 | Pentose phosphate pathway | 46 | 0.196581197 | 0.000666312 | 3.176322631 | 0.003203421 |
| path:ath00053 | Ascorbate and aldarate metabolism | 39 | 0.276595745 | 5.08E−07 | 6.293861872 | 4.24E−06 |
| path:ath00860 | Porphyrin and chlorophyll metabolism | 37 | 0.26618705 | 2.70E−06 | 5.568891264 | 2.11E−05 |
| path:ath00280 | Valine, leucine and isoleucine degradation | 37 | 0.185 | 0.006096775 | 2.214899808 | 0.021774198 |
| path:ath00061 | Fatty acid biosynthesis | 37 | 0.183168317 | 0.007176271 | 2.14410114 | 0.024917609 |
| path:ath00592 | alpha-Linolenic acid metabolism | 36 | 0.204545455 | 0.001176924 | 2.929251557 | 0.005254125 |
| path:ath00220 | Arginine biosynthesis | 28 | 0.24137931 | 0.000264729 | 3.577197832 | 0.001438747 |
| path:ath00640 | Propanoate metabolism | 27 | 0.197080292 | 0.007622503 | 2.117902407 | 0.025751699 |
| path:ath00670 | One carbon pool by folate | 20 | 0.298507463 | 9.32E−05 | 4.030622411 | 0.000529499 |
| path:ath00290 | Valine, leucine and isoleucine biosynthesis | 18 | 0.25 | 0.002061633 | 2.685788684 | 0.008053253 |
| path:ath00196 | Photosynthesis | 14 | 0.4375 | 8.45E−06 | 5.073151505 | 5.87E−05 |
| path:ath00650 | Butanoate metabolism | 14 | 0.259259259 | 0.004375359 | 2.358986273 | 0.016573331 |
| path:ath00300 | Lysine biosynthesis | 10 | 0.37037037 | 0.000836778 | 3.077389825 | 0.003873972 |
| path:ath00261 | Monobactam biosynthesis | 9 | 0.409090909 | 0.000654796 | 3.183893939 | 0.003410396 |

were obtained through selecting nodes with a high betweenness centrality (BC) value (BC value >0.02) or a large degree (D) value (D value >10). In P2 *versus* P1, 30 nodes had a high BC value, 29 nodes had a large degree value, and 14 nodes had both a large BC and degree

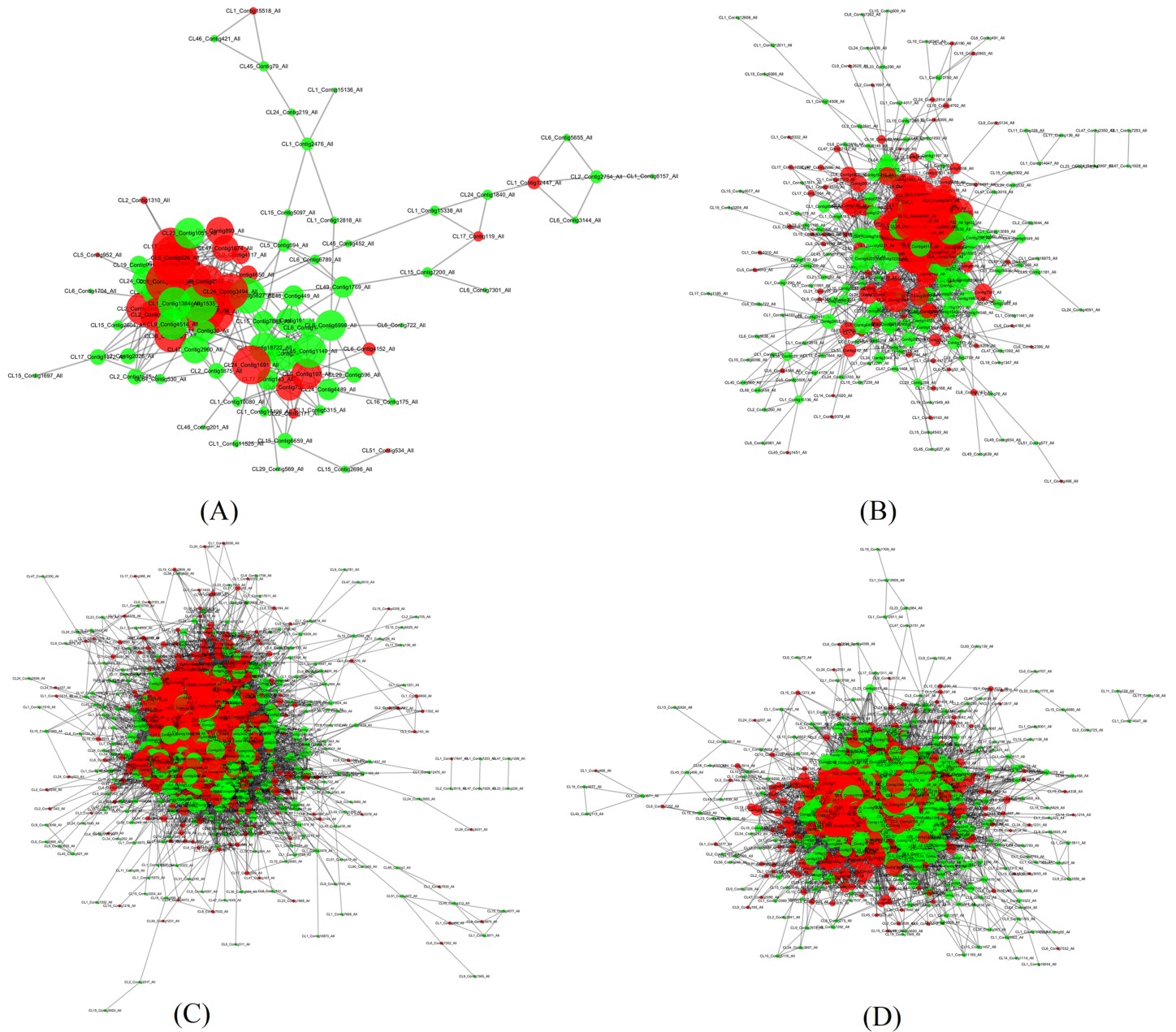

**Figure 6 Illustration of the PPI network (A) The network of P2 *versus* P1, (B) The network of P3 *versus* P1, (C) The network of P4 *versus* P1, (D) The network of P5 *versus* P1. Network nodes represent proteins. Edges represent protein-protein associations. The green nodes.**

value; in P3 *versus* P1, 15 nodes had a high BC value, 118 had a large degree value, and 15 nodes had both a large BC and degree value; in P5 *versus* P1, 15 nodes had a high BC value, 592 nodes had a large degree value, and seven nodes had both a large BC and degree value; and in P5 *versus* P1, 439 nodes had a high BC value, 11 nodes had a large degree value, and 10 nodes had both a large BC and degree value (Table 3). Among the nodes with both a large degree and high BC value, TPI was shared in all four pairs, AT3G29320 was shared between three pairs (P2 *versus* P1, P3 *versus* P1, P4 *versus* P1), CDC5 was shared between

**Table 3 The list of nodes with both a high BC value (>0.02) and high degree value (>10).**

| Pairs | Gene | BC value | D value |
|---|---|---|---|
| P2 *versus* P1 | TPI[a] | 0.1870093 | 19 |
| | AT1G11860 | 0.17621229 | 17 |
| | AT3G29320[b] | 0.12413995 | 13 |
| | NRPB2[d] | 0.098109 | 17 |
| | emb1473 | 0.09009896 | 25 |
| | ACP4 | 0.08761633 | 18 |
| | P5CS2[f] | 0.04926569 | 13 |
| | NDPK2 | 0.04680682 | 13 |
| | AT1G12230 | 0.0442739 | 15 |
| | LOS2 | 0.04314511 | 15 |
| | rps15 | 0.04216912 | 16 |
| | AT2G43030 | 0.03732062 | 24 |
| | PP2AA2 | 0.03569743 | 11 |
| | GAPC1[d] | 0.03315947 | 13 |
| P3 *versus* P1 | CDC5[c] | 0.14663404 | 40 |
| | Hsp70b | 0.10052988 | 38 |
| | TPI[a] | 0.05953945 | 41 |
| | PSP | 0.04886272 | 16 |
| | AT3G29320[b] | 0.0481939 | 21 |
| | NRPB2[d] | 0.04459201 | 41 |
| | GS2[e] | 0.04413029 | 25 |
| | CPN10 | 0.04098666 | 39 |
| | LOX2 | 0.03694796 | 14 |
| | mtLPD1 | 0.03669113 | 24 |
| | AT1G09640 | 0.0346852 | 55 |
| | CDPMEK | 0.03296113 | 18 |
| | GAPC1[d] | 0.03215865 | 26 |
| | PUR5 | 0.0311079 | 38 |
| P4 *versus* P1 | CDC5[c] | 0.04350023 | 125 |
| | TPI[a] | 0.03028585 | 160 |
| | AT3G29320[b] | 0.02679787 | 23 |
| | P5CS2[f] | 0.02186006 | 118 |
| | HSP70 | 0.02005322 | 158 |
| P5 *versus* P1 | CDC5[c] | 0.04774593 | 87 |
| | AT5g06290 | 0.03391341 | 118 |
| | TPI[a] | 0.02928484 | 145 |
| | HEME2 | 0.02682115 | 124 |
| | HSC70-1 | 0.0264096 | 84 |
| | AT3G54470 | 0.02636206 | 102 |
| | AT5G51970 | 0.02472014 | 67 |
| | GS2[e] | 0.02222725 | 70 |

**Notes:**
[a] Gene shared in all pairs.
[b] Gene shared in P2 *versus* P1, P3 *versus* P1, P4 *versus* P1.
[c] Gene shared in P3 *versus* P1, P4 *versus* P1, P5 *versus* P1.
[d] Gene shared in P2 *versus* P1, P3 *versus* P1.
[e] Gene shared in P3 *versus* P1 and P5 *versus* P1.
[f] Gene shared in P2 *versus* P1 and P4 *versus* P1.

three pairs (P3 *versus* P1, P4 *versus* P1, P5 *versus* P1), TP1 was shared between three pairs (P2 *versus* P1, P3 *versus* P1, P4 *versus* P1), NRPB2 and GAPC1 were shared between two pairs (P2 *versus* P1, P3 *versus* P1), GS2 was shared between two pairs (P3 *versus* P1, P5 *versus* P1), and P5CS2 was shared between two pairs (P2 *versus* P1, P4 *versus* P1). TPI is a protein with both the highest BC value and CC value, and emb1473 is a protein with the largest degree in the network of P2 *versus* P1. TPI had a degree value of 19, and occupied the central position in the network because of its high degree, BC, and CC values. TPI was also considered to be centrally located in the network of P4 *versus* P1, and P5 *versus* P1 due to its high degree, BC, and CC values in those networks. In the network of P3 *versus* P1, the RPL4 protein encoded by AT5G02870 had the largest degree value, the CDC5 protein had the highest BC value, and the Hsp70b protein had the highest CC value. In the P3 *versus* P1 network, the CDC5 protein had a degree value of 40 and a CC value of 0.41441441, and occupied the central position. In the P4 *versus* P1 network, the PRPL3 protein encoded by AT2G43030 had degree value of 184, the largest in the network. The CDC5 protein had the highest BC value in P4 *versus* P1, and in P5 *versus* P1. These results indicate that these proteins play a vital role in these large networks.

## Identification of differential metabolites

Based on the results of the QC and QA analyses (Fig. S33), all samples exhibited a high quality, and could be used for subsequent screening and identification of differential metabolites (DMs). According to the PCA, the components of the five samples (P1, P2, P3, P4, P5) displayed effective separation (Figs. 7A and 7B). As a supervised method, a PLS-DA (partial least squares discriminant analysis), the most commonly used classification method in metabonomics, was performed to confirm the PCA results. PLS-DA also has potential applications in sample classification. Satisfactory modeling and prediction results were obtained from all sample comparison groups despite low Q2 values, suggesting metabolomes are distinguishable under water-deficit conditions (Figs. 8A–8D). In addition, the OPLS-DA (orthogonal partial least squares-discriminant analysis) showed a remarkable separation among the five samples (Figs. 7C and 7D). Furthermore, based on the parameter VIP (variable of importance in prediction) >1, which is a measure of the variable importance in the OPLS-DA, a total of 1,422 differential metabolites (DMs) were identified in all five samples, which were displayed as a heat map (Fig. 9). The detailed information of the DMs from this 6-sample group comparisons (P2 *versus* P1, P3 *versus* P1, P4 *versus* P1, P5 *versus* P1, P3 *versus* P2, and P4 *versus* P2) are shown in Tables S3–S8. The number of DMs in the P4 *versus* P1 group was the highest, whereas the P3 *versus* P2 group had the lowest number of DMs, which is similar to the DEP results (Fig. 7E). More up-regulated DMs were identified through metabolomic analysis compared to the DEPs identified from the same samples through RNA-seq based transcript profiling.

The metabolite levels of five comparison groups are shown in Figure S34. A total of five metabolites, including Ubiquinone-1, perillyl alcohol, phosphoserine, d-Limonene, and 2-Amino-2-dexy-D-gluconate, exhibited higher levels in samples under water withholding conditions compared to control.

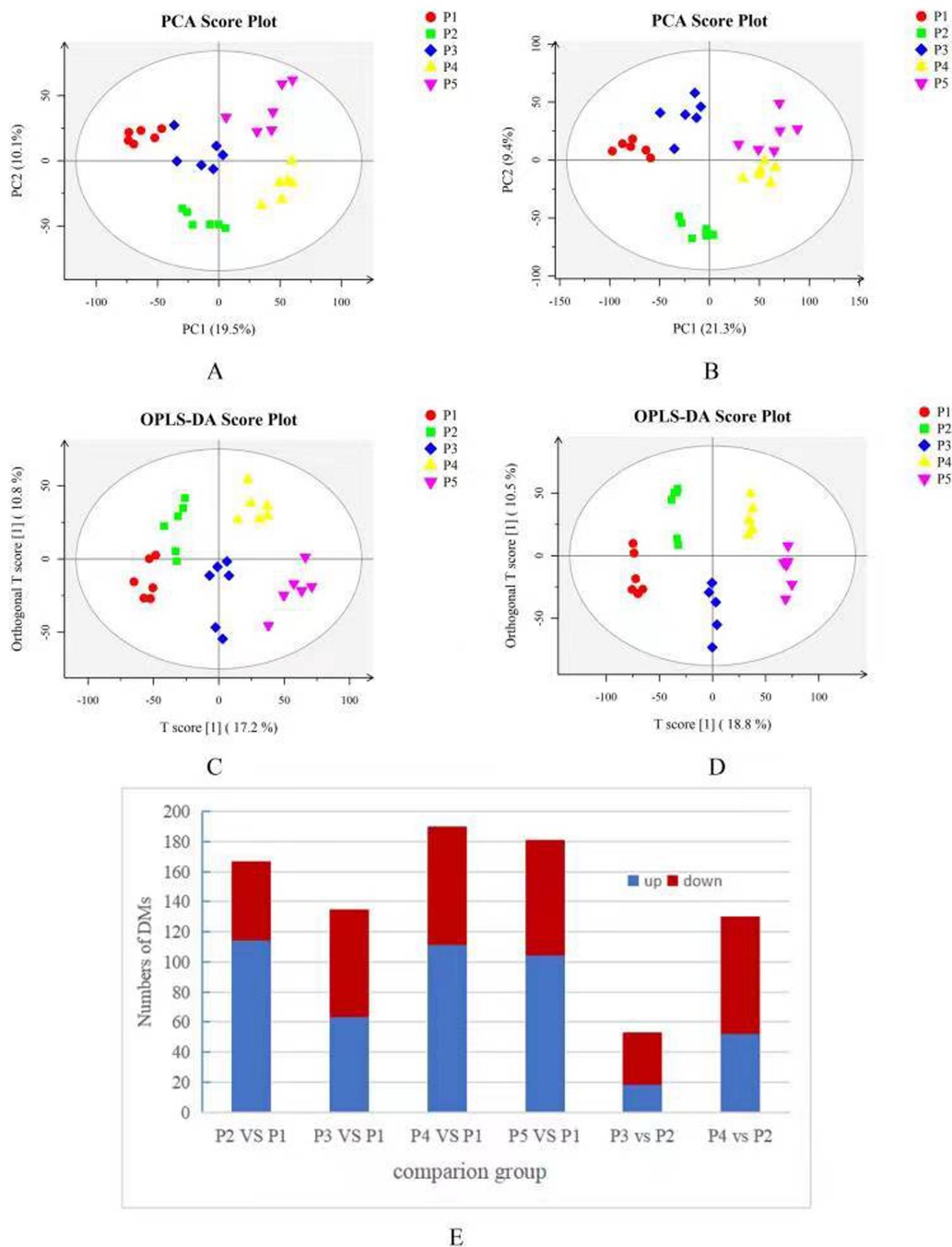

**Figure 7** **(A) PCA score plot in positive ion mode. (B) PCA score plot in negative ion mode. (C) OPLS-DA score plot in positive ion mode. (D) OPLS-DA score plot in negative ion mode. (E) 10 Statistics of DMs from six sample-pairs, being namely P2 *versus* P1, P3 *versus* P1, P3 *versus* P2, P4 *versus* P1, P4 *versus* P2, P5 *versus* P1. Red: upregulated expressed proteins, blue: downregulated expressed proteins.**

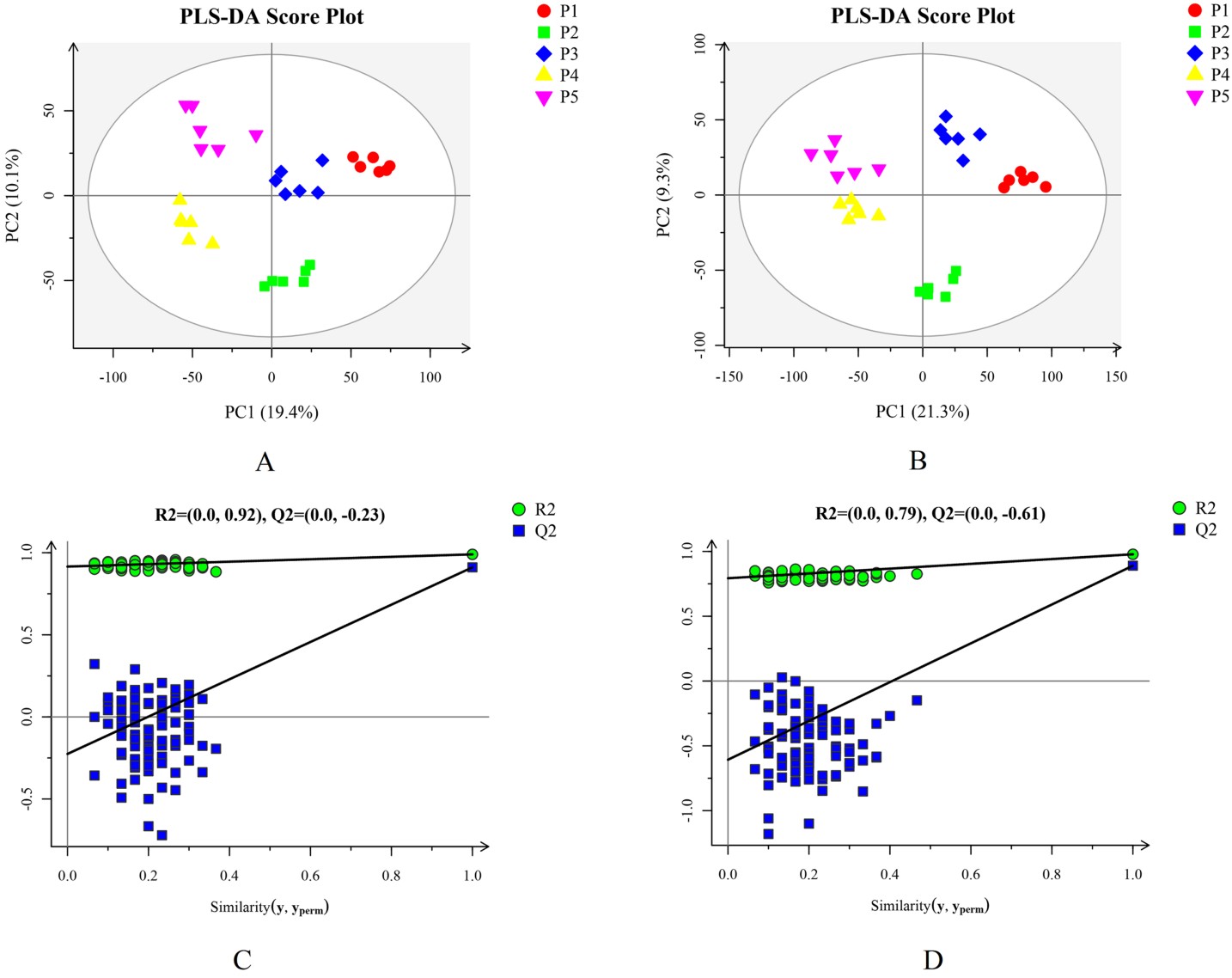

**Figure 8 Plots of PLS-DA score and permutation test. (A) PLS-DA score plot in positive ion mode. (B) PLS-DA score plot in negative ion mode. (C) OPLS-DA permutation test plot in positive ion mode. (D) OPLS-DA permutation test plot in negative ion mode.**

## KEGG pathways of DMs

A total of 331 DMs from all five samples were identified in the KEGG database: 95 DMs were found in P2 *versus* P1, 60 DMs in P3 *versus* P1, 172 DMs in P4 *versus* P1, 135 DMs in P5 *versus* P1, 53 DMs in P3 *versus* P2, and 82 DMs in P4 *versus* P2 (Tables S9–S14). There were 22, 11, 41, 32, 1, and 9 up-regulated DMs with a fold change >5 in each pair, respectively. Among them, ubiquinone-1 and xanthoxic acid were shared in P2 *versus* P1, P3 *versus* P1, P4 *versus* P1, and P5 *versus* P1, and L-isoleucine was shared in P3 *versus* P2 and P4 *versus* P2. Some DMs were only found in the samples under water-deficient conditions. Dimethyl sulfone was unique to the samples after 5 days (P2), 15 days (P4),

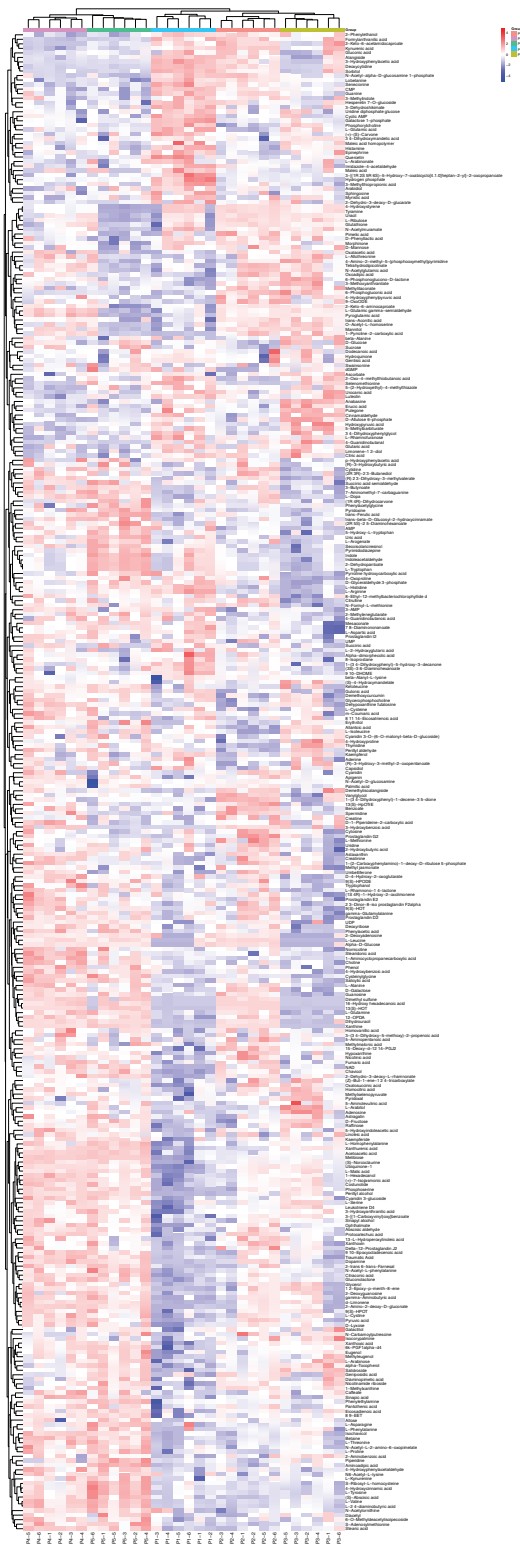

**Figure 9 The heat map of differential metabolite among five samples.** The columns represent samples, the rows represent metabolites, different colors indicate the relative content of the differential metabolites.

and 20 days (P5) of water withholding. Xanthine, dihydrouracil, and 13(S)-HOT were only observed in the P2 and P4 samples. The accumulation of some DMs was reduced in samples under conditions of water deficiency compared to controls, including: 3-methylthiopropionic acid, cyclic AMP, 3-dehydroshikimate, L-arginine, CMP, 3-hydroxyphenylacetic acid, galactose 1-phosphate, and deoxycytidine. Based on the pathway enrichment assessment, tyrosine metabolism was the only significantly enriched pathway (FDR < 0.05, pathway impact values ≥0.2) in the P5 *versus* P1 group comparison (Table S12). The only significantly enriched pathway in the P3 *versus* P2 group comparison was arginine and proline metabolism (Table S13), whereas the rest of group comparisons had no significantly enriched pathways. Tyrosine metabolism includes nine components: 6 up-regulated DMs (3,4-dihydroxyphenylethyleneglycol, 3,4-dihydroxy-L-henylalanine, L-tyrosine, succinate semialdehyde, dopamine, fumarate, acetoacetate, 4-hydroxy-phenylacetaldehyde) and one down-regulated DM (tyramine; Table S12). A total of 11 components were linked to arginine and proline metabolism: glyoxylate, L-ornithine, L-glutamate, L-proline, L-1-pyrroline-3-hydroxy-5- carboxylate, S-adenosyl-L-methionine, L-arginine, pyruvate, hydroxyproline, (4R)-4-hydroxy-2-oxoglutarate, and spermidine (Table S13). Furthermore, a correlation analysis of the differential metabolites showed that ubiquinone-1 accumulation was the most positively correlated to L-tyrosine accumulation, and xanthoxic acid content was positively correlated with ubiquinone-1 and L-tyrosine in the P5 *versus* P1 group. L-tyrosine composition in the P5 *versus* P1 group was also negatively correlated with cyclic AMP, 3-dehydroshikimate, CMP, 3-hydroxypheny-lacetic acid, and deoxycytidine. In the P3 *versus* P2 group, a significant positive correlation was found between L-proline and L-isoleucine accumulation. As a marked osmotic modulation in response to drought stress, the proline accumulation in each sample was investigated more closely. L-proline showed a significant accumulation after 20 days of water withholding (P5) compared with control (P1; Table S6) and 4-hydroxyproline content was increased in P3 *versus* P1, and in P4 *versus* P1 (Tables S4 and S5).
The concentration of 4-hydroxyproline and L-proline were both increased in P3 *versus* P2, and in P4 *versus* P2 (Tables S7 and S8).

## DISCUSSION

### The decline of photosynthesis and glycometabolism-related proteins and metabolites resulting in water stress

Water stress affects protein biosynthesis and degradation, and the photosynthetic process (*Amin et al., 2009*). Similar to those found in wheat (*Michaletti et al., 2018*), some photosynthetic-related proteins, mainly photosystem II oxygen-evolving enhancer protein 1, photosystem I reaction center subunit (psaK), photosystem II Psb27 protein, and ribulose-bisphosphate carboxylase, were down-regulated in samples under water deficiency conditions. The expression patterns of these proteins were confirmed in our study by the significant reduction of sorbitol observed during water stress, which is the main end-product of photosynthesis, and is essential for stamen development in apple trees (*Meng et al., 2018*). However, the levels of photosystem I subunit IV and photosystem
II oxygen-evolving enhancer protein 2 increased in water deficit conditions, implying these substrates, components of the photosynthetic system, exhibit different roles in response to photosynthesis impairment induced by drought stress. As reported in the water-stressed leaves of apple trees (*Yang et al., 2019*), a significant reduction in photosynthesis is generally correlated with changes in sugar metabolism.

In this study, six of the top KEGG pathways connected to glycometabolism were also influenced by water stress, including glycolysis/gluconeogenesis, pyruvate metabolism, glyoxylate and dicarboxylate metabolism, citrate cycle (TCA cycle), fructose and mannose metabolism, and the pentose phosphate pathway. Declines were mainly seen in ribose 5-phosphate isomerase A, alpha-N-acetylglucosaminidase, pyruvate dehydrogenase E1 component alpha subunit (EC:1.2.4.1), triosephosphate isomerase (TPI), and glycosyltransferase (AT3G29320). TPI occupied the central position in both the P4 *versus* P1 network and the P5 *versus* P1 network due to its high degree, BB, and CC values, as it plays an important role in the glycolysis pathway. A recent study in *Barley* indicated TPI could be linked to drought tolerance in a comparative proteome-transcriptome analysis (*Wójcik-Jagła et al., 2020*). However, the expression pattern of the TPI protein was not consistent with the direction of changes seen in transcript accumulation during water stress. This could be partly due to the instability of transcripts, which are prone to RNAse degradation (*Wójcik-Jagła et al., 2020*). For the glycosyltransferase gene, the pattern of changes in protein and transcript accumulation was very similar under water stress conditions. *Zheng et al. (2017)* confirmed that QUA1, which has been identified as a glycosyltransferase in *Arabidopsis*, increases drought tolerance by regulating chloroplast-associated calcium signaling. Similar findings have also been shown in rice (*Oryza sativa* L.; *Dinesh et al., 2017*). The following pathways involved in carbohydrate metabolism were found to be down-regulated in our KEGG-based metabonomics analyses despite high FDR values: fructose and mannose metabolism, glyoxylate and dicarboxylate metabolism, pyruvate metabolism, glycolysis/gluconeogenesis, pentose phosphate pathway, and starch and sucrose metabolism. These results are not only consistent with our proteome data, but also match the results of previous transcriptome analyses (*Shi et al., 2020*). Similar results have also been observed in other plants, such as *Medicago truncatula* (*Lyon et al., 2016*) and spring-wheat (*Michaletti et al., 2018*).

## The disturbance of amino acid metabolism was induced by water stress

Most DEPs identified in this study were enriched in "biosynthesis of secondary metabolites" and "biosynthesis of amino acids," which is consistent with previous RNA-seq results (*Shi et al., 2020*). However, only "tyrosine metabolism" and "arginine and proline metabolism" were considered significantly enriched pathways in our metabolomic analysis. It is well known that secondary metabolism is critical to plant growth and development, and can be induced by both biotic and abiotic stresses (*Fox et al., 2017*). The involvement of secondary metabolites in response to drought stress is extremely complicated and depends on various parameters, such as high temperature and photoinhibition, which typically accompany drought stress (*Niinemets & Way, 2016*).

Previous studies have demonstrated that water deficiency can damage the biosynthesis of secondary metabolites in plants, interfering with normal growth and generating chlorosis, which reduces plant production or even causes the plant to die (*Afshar, Gürbüz & Uyanik, 2012*; *Bitarafan et al., 2019*). Transcriptomic analyses have shown that secondary metabolism in plants is regulated by a large number of transcription factors, most of which belong to the bHLH, MYB, MYB-like, C2H2, and bZIP families and are down-regulated during water stress (*Shi et al., 2020*). The down-regulation of MYB-related transcription factor LHY (MYB-like families) and transcription factor MYC2 (bHLH families) in drought conditions have been further confirmed using proteomic approaches, suggesting that they might be pivotal candidate genes for subsequent verification.

This study also found that the reduction in proteins linked to secondary metabolites mainly involved NADH-dependent glutamate synthase 1 isoform 1 (K00264 glutamate synthase (NADPH/NADH) (EC:1.4.1.13 1.4.1.14)), lipoxygenase (K00454 lipoxygenase (EC:1.13.11.12)), allene oxide synthase, (K01723 hydroperoxide dehydratase (EC:4.2.1.92)), and the peroxidase superfamily protein (K00430 peroxidase (EC:1.11.1.7)). The genes corresponding to these proteins all had reduced expression levels except the peroxidase superfamily protein. Four genes related to glutamate synthase (NADPH/NADH) (EC:1.4.1.13 1.4.1.14) were down-regulated in the water shortage samples (P5); this down-regulation aligned with the reduction of L-glutamic content during water stress. The NADH-dependent glutamate synthase (NADH-GOGAT), which uses NADH as the electron donor, is present mostly in non-photosynthesizing cells, where the reductant is supplied by the pentose phosphate pathway (*Forde & Lea, 2007*). The importance of NADH-GOGAT in ammonium assimilation has previously been reported in various species (*Konishi et al., 2014*), as well as its potential links to drought response through amino acid metabolism. It has been demonstrated that disruptions in the amino acid metabolism of plants can be attributed to decreases in NADH-GOGAT activity (*Forde & Lea, 2007*). A special regulation mode of amino acid metabolism associated with drought stress tolerance has been reported in wheat (*Aidoo et al., 2017*), *Lotus japonicus* (*Sanchez et al., 2012*), and maize plants (*Alvarez et al., 2008*). A total of 20 types of amino acids were obtained in our metabonomics analysis (Table S15). The changes in the patterns of the different amino acids varied under different water stress conditions, similar to the changes observed in the *Lotus japonicus* species (*Sanchez et al., 2012*) and in maize plants (*Alvarez et al., 2008*). Phosphoserine content increased in all water shortage samples compared with control, and the concentration of both L-arginine and L-glutamic acid decreased. Notably, the L-proline content, which is a well-known bio-marker for water deficit, was significantly higher in the sample after 20 days of water withholding (P5) compared with control (P1), but this increase was not observed in the other water shortage samples. An accumulation of 4-hydroxyproline was observed in P3 *versus* P1, and in P4 *versus* P1, while the arginine and proline metabolism pathway, which involves 11 components, was only enriched in P3 *versus* P2. Among these 11 components, L-proline and 4-hydroxyproline amounts were increased in samples after 7 days of water withholding (P3). Proline changes are associated with extreme water scarcity in many plant species (*Witt et al., 2012*; *Pirzad et al., 2011*), but these changes are genotype specific and also related to the extent of the water stress

(*Bowne et al., 2012*). Proline is known as a compatible solute essential for osmotic adjustments. It protects cellular structures during water stress and also plays an important role in ROS (reactive oxygen species) scavenging (*Zadebagheri, Azarpanah & Javanmardi, 2014*), thus alleviating the adverse effects of drought stress on plant metabolism. It is thus reasonable to conclude that disturbances in the amino acid metabolism observed in this study was due to the enhanced protein breakdown induced by corresponding down-regulated genes.

## The tyrosine-derived pathway is important for drought tolerance

Our results highlight the importance of tyrosine metabolism, which was a unique significantly enriched pathway in the comparison of water stress conditions (P5) and control (P1) in our study. As a key enzyme in the tyrosine-derived pathway, tyrosine aminotransferase (TAT) catalyzes the reversible interconversion of tyrosine and 4-hydroxyphenylpyruvate for the biosynthesis of secondary metabolites. According to a previous transcriptome analysis (*Shi et al., 2020*), the TAT gene is up-regulated during water deficit, which is in agreement with the corresponding enzyme in our proteomic analysis. A recent study in apple trees (*Malus domestica*) found the same accumulation pattern of ubiquinone-1 in the metabolome, reinforcing the hypothesis that TAT genes confer drought tolerance (*Wang et al., 2018a*). Ubiquinone (UQ) is an important prenyl quinone whose core cyclic scaffold is provided by the tyrosine-derived pathway. UQ functions as an electron transporter in the respiratory chain and is indispensable in a plant's response to abiotic stress (*Liu & Lu, 2016*). We found a significant accumulation of dopamine after 20 days of water withholding compared to control. Some studies have reported that dopamine confers drought tolerance in plants. According to a correlation analysis of metabolites, the contents of dopamine and ubiquinone-1 were all significantly positively correlated with L-tyrosine accumulation, implying that okra plants could improve resistance to drought and prevent drought-induced damage by enhancing tyrosine metabolism and its derivatives.

We observed significant decreases in the abundance of glutamine synthetase (GS) proteins after 7 days and 20 days of water withholding. The corresponding GS gene was also down-regulated in water stress samples. The decline of L-glutamic acid during water stress was consistent with the expression pattern of the GS gene and protein. GS2 was also shown to be important in protein interaction networks because of its large degree and high BC values. The reduction of L-glutamic acid observed could be linked to tyrosine metabolism accumulation and the synthesis of arginine. Similar findings have been reported in the metabolome of wheat (*Michaletti et al., 2018*). Glutamic acid (Glu) can supply amino groups for photorespiratory metabolism, and also ornithine to produce arginine (Arg) for carbon (C) and nitrogen (N) assimilation and partitioning (*Díaz et al., 2005*). GS is also known as a metabolic indicator for drought stress tolerance in wheat (*Nagy et al., 2013*), which is further supported by previous studies of GS protein abundance in many plant species (*Wang et al., 2018b*). Water stress conditions affects the balance between photosynthetic carbon uptake and the use of photoassimilates, causing alterations in the sugar pools (*Michaletti et al., 2018*). This further supports the hypothesis that
tyrosine metabolism could confer drought tolerance to plants by influencing carbon and nitrogen metabolism. Further research should focus on the regulation mechanism of the GS2-mediated protein interaction network in the response of okra plants to drought stress.

## CONCLUSION

Comparing transcriptomic, proteomic, and metabolomic data showed an obvious connection between all three, especially the metabolome and proteome. Water stress disrupts the biosynthesis of secondary metabolites, especially in amino acid metabolism, which is associated with the inhibition of photosynthesis and glycometabolism. The components of the tyrosine-derived pathway play key roles in improving drought tolerance in okra plants.

### Funding

This study was funded by the Guizhou Fundamental Research Program (Natural Science Project) under grant number QianKeHeJiChu-ZK[2022]YiBan006 and the Special Fund for Guiyang College supported by Guiyang Science and Technology Bureau (GYU-KY-[2022]). The funders had no role in study design, data collection and analysis, decision to publish, or preparation of the manuscript.

### Grant Disclosures

The following grant information was disclosed by the authors:
Guizhou Fundamental Research Program Natural Science Project: QianKeHeJiChu-ZK [2022]YiBan006.
Guiyang Science and Technology Bureau: GYU-KY-[2022].

### Competing Interests

The authors declare that they have no competing interests.

### Author Contributions

- Jiyue Wang conceived and designed the experiments, performed the experiments, analyzed the data, authored or reviewed drafts of the article, and approved the final draft.
- Denghong Shi performed the experiments, analyzed the data, authored or reviewed drafts of the article, and approved the final draft.
- Yu Bai analyzed the data, authored or reviewed drafts of the article, and approved the final draft.
- Ting Zhang performed the experiments, prepared figures and/or tables, and approved the final draft.
- Yan Wu performed the experiments, prepared figures and/or tables, and approved the final draft.
- Zhenghong Liu performed the experiments, prepared figures and/or tables, and approved the final draft.

- Lian Jiang performed the experiments, prepared figures and/or tables, and approved the final draft.
- Lin Ye performed the experiments, prepared figures and/or tables, and approved the final draft.
- Zele Peng performed the experiments, prepared figures and/or tables, and approved the final draft.
- Hui Yuan performed the experiments, prepared figures and/or tables, and approved the final draft.
- Yan Liu conceived and designed the experiments, authored or reviewed drafts of the article, and approved the final draft.

## Data Availability

The raw data is available in the Supplemental Files.

## Supplemental Information

Supplemental information for this article can be found online at http://dx.doi.org/10.7717/peerj.14312#supplemental-information.

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
