# Peer review of "Comprehensive proteomic and metabolomic analysis uncover the response of okra to drought stress"

_PeerJ, doi:10.7717/peerj.14312_

## Round 0.1 · original submission · Major Revisions

The manuscript needs a thorough revision.

·

Basic reporting

no comment

Experimental design

no comment

Validity of the findings

no comment

Additional comments

In this study, comprehensive proteomic combined with metabolome were used to analyze the changes of proteins, and metabolites in okra leaf under different degrees of water-stress. Their results showed that the photosynthesis and glycometabolism in okra leaf were both adversely affect by drought stress. Tyrosine metabolism can enhance the tolerance to drought stress by affecting carbon and nitrogen metabolism. The results of the article are messy and the language is not logical. The expression of English language should be improved to ensure that the contents can be clearly understand. Some specific comments can be found below.

1. Abstract section: “transcriptomics” should be “Proteomic” in line 41? Did the authors performed transcriptome? I don't see transcriptome data.

2. Introduction section: “Pravisya et al., (2018) reported that Pseudomonas fluorescens (PF) was responsible for effectively mitigating drought stress in okra plants” in line 75-76, which is irrelevant to the topic of this article. It is suggested to increase the application of multiomics in drought stress.

3. Materials and Methods section: In line 99, “dehydration treatment was applied to all plants”. How does drought treatment was performed? Please describe it more detail.

4. Materials and Methods section: “RNA” should be “protein” in line 102? TMT and iTRAQ are two different labeling methods, please determine which method was used and keep it consistent with the text.

5. Results section: “iTRAQ” should be “TMT” in line 239? “1,809 subgroups of biological process” in line 273, are there so many GO terms? Please confirm. “AT5G02870” and “AT2G43030”, please confirm whether it is correct. “RNA-seq” in line 372, please confirm.

6. Figure legend should be rewritten. Especially, Fig.1 and Fig.3. Similar graphs can be merged into one graph, such as Figure 7, 9 and 10. There are many supplementary data, some of which can be put right in the figure.

7. The format of the article should meet the requirements of the Journal. Font size, punctuation marks, etc. should be in English style. The format of references should be consistent, such as full name or abbreviation of the journal.

·

Basic reporting

1. English Language as well as grammar needs improvement throughout the manuscript.
2.

Experimental design

The submitted manuscript is aligned to scope of the journal. My observations are
1. Please explain how the applied stress was quantified? Whether any data was recorded on soil moisture content/water potential.
2. It would have been better if the response of drought susceptible okra genotype would have been compared to the tolerant genotype.
3. Include the soil properties as well.
4. How the light intensity was maintained in incubator ?
5. Since the experiment was conducted in constant temperature incubator, it does not mimic the field conditions. How the results can be extrapolated to true responses?

Validity of the findings

In silico analysis has been performed satisfactorily. Discussion needs major improvement. My comments are as follows:
1.Explain L423-424 in light of sorbitol accumulation in apple leaves in response to drought (https://doi.org/10.1016/j.plaphy.2019.05.025)
2. L429-431 explain more clearly.
3.L449-450 Explain clearly which pathways?
4.L451-L454. Explain clearly taking reference into consideration.
5. L463: Explain the parameters.
6. L463-465: No references provided.
7. L485: Name the species.
8. L540-542: Rewrite the section
9. Conclusion needs to more specific and result oriented.

---

## Round 0.2 · Minor Revisions

The manuscript is improved substantially but still needs some final language polishing before Acceptance.

·

Basic reporting

Please check your work for spelling and punctuation.

Experimental design

no comment

Validity of the findings

no comment

---

## Round 0.3 · accepted · Accept

The paper is revised to a satisfactory level

·

Basic reporting

Article meets the suitability criterion after revision.

Experimental design

Article meets the suitability criterion after revision.

Validity of the findings

Article meets the suitability criterion after revision.